# Fermented Milk Supplemented with Sodium Butyrate and Inulin: Physicochemical Characterization and Probiotic Viability Under In Vitro Simulated Gastrointestinal Digestion

**DOI:** 10.3390/nu17132249

**Published:** 2025-07-07

**Authors:** Katarzyna Szajnar, Małgorzata Pawlos, Magdalena Kowalczyk, Julita Drobniak, Agata Znamirowska-Piotrowska

**Affiliations:** 1Department of Dairy Technology, Faculty of Technology and Life Sciences, University of Rzeszów, Ćwiklińskiej 2D, 35-601 Rzeszów, Poland; mpawlos@ur.edu.pl (M.P.); mkowalczyk@ur.edu.pl (M.K.); aznamirowska@ur.edu.pl (A.Z.-P.); 2Student Scientific Association “Ferment”, Faculty of Technology and Life Sciences, University of Rzeszów, Ćwiklińskiej 2D, 35-601 Rzeszów, Poland; jd119077@stud.ur.edu.pl

**Keywords:** probiotic bacteria, inulin, sodium butyrate, in vitro digestion

## Abstract

**Background/Objectives:** Probiotics are increasingly recognized for their role in managing gastrointestinal disorders through modulation of gut microbiota. Restoring microbial balance remains a therapeutic challenge. Recent strategies combine probiotics, inulin, and sodium butyrate as synergistic agents for gut health. This study aimed to evaluate the effects of milk supplementation with inulin and sodium butyrate on physicochemical properties, sensory characteristics, and the survival of selected probiotic strains during in vitro simulated gastrointestinal digestion. **Methods:** Fermented milk samples were analyzed for color, pH, titratable acidity, and syneresis. A trained sensory panel evaluated aroma, texture, and acceptability. Samples underwent a standardized in vitro digestion simulating oral, gastric, and intestinal phases. Viable probiotic cells were counted before digestion and at each stage, and survival rates were calculated. **Results:** Physicochemical and sensory attributes varied depending on probiotic strain and supplementation. Inulin and the inulin–sodium butyrate combination influenced syneresis and acidity. *Lacticaseibacillus casei* 431 and *Lactobacillus johnsonii* LJ samples showed the highest viable counts before digestion. Two-way ANOVA confirmed that probiotic strain, supplementation type, and their interactions significantly affected bacterial survival during digestion (*p* < 0.05). **Conclusions:** The addition of inulin and sodium butyrate did not impair probiotic viability under simulated gastrointestinal conditions. The effects on product characteristics were strain-dependent (*Bifidobacterium animalis* subsp. *lactis* BB-12, *L. casei* 431, *L. paracasei* L26, *L. acidophilus* LA-5, *L. johnsonii* LJ). These findings support the use of inulin–butyrate fortification in dairy matrices to enhance the functional potential of probiotic foods targeting gut health.

## 1. Introduction

Gastrointestinal disorders present escalating therapeutic challenges, particularly in the context of restoring intestinal microbiota. In this regard, sodium butyrate, inulin, and probiotics have emerged as key agents in contemporary dietary strategies. Probiotics play a crucial role in the management of irritable bowel syndrome (IBS) and various gastrointestinal pathologies, owing to their capacity to modulate gut microbiota composition, enhance epithelial barrier integrity, and regulate host inflammatory responses. Clinically relevant species include *Lacticaseibacillus rhamnosus* GG, *Lacticaseibacillus casei*, *Lactobacillus acidophilus*, *Bifidobacterium animalis* ssp. *lactis* BB-12, and *Lactobacillus johnsonii*. However, probiotic efficacy is critically contingent on survival during gastrointestinal transit and interaction with bioactive food components.

Butyric acid, a major short-chain fatty acid produced by gut microbes, serves as the preferred energy substrate for colonocytes and reinforces epithelial barrier integrity by upregulating mucin production and tight-junction proteins (e.g., occludin, claudin-1) in vivo and in vitro [1,2]. Beyond its trophic effects, butyrate acts as a potent immunomodulator: it activates G-protein-coupled receptors such as GPR109A and inhibits histone deacetylases, thereby promoting the differentiation of Foxp3^+^ regulatory T cells and increasing interleukin-10 secretion [3]. Importantly, physiological concentrations of sodium butyrate do not inhibit the growth of lactic acid bacteria, including *Lactiplantibacillus plantarum*, under standard culture conditions [4]. In fermented and synbiotic formulations, metabolic cross-feeding between lactate-producing probiotics (e.g., *Lactobacillus*, *Bifidobacterium*) and butyrate-producing commensals such as *Faecalibacterium prausnitzii* substantially amplifies luminal butyrate levels and supports co-colonization of these beneficial microbes [5]. Moreover, butyrate supplementation enhances probiotic resilience during simulated gastrointestinal transit, likely via induction of stress-response systems (e.g., bile salt hydrolases, F_0_F_1_-ATPases) and provision of fermentable substrates [6]. Collectively, these synergistic mechanisms underlie the improved survival, colonization, and functional efficacy of probiotic strains in butyrate-fortified fermented foods. Although small amounts of sodium butyrate are present in dairy and fermented foods, dietary intake rarely reaches levels sufficient to exert trophic effects on the intestinal epithelium. Co-administration of sodium butyrate with probiotic strains offers promising therapeutic avenues—particularly in IBS and antibiotic-associated complications—by potentiating probiotic and prebiotic efficacy.

Butyrate enhances intestinal barrier function by upregulating the expression of tight-junction proteins (e.g., occludin, zonulin, and claudins), stimulating mucin (MUC2) secretion, and promoting the synthesis of antimicrobial peptides (AMPs) such as LL-37, RegIIIγ, and β-defensins. Through these actions, butyrate contributes to barrier homeostasis and attenuates mucosal inflammation, thereby creating a more favorable niche for probiotic colonization and proliferation [7]. Beyond its nutritive role, butyrate acts as a signaling mediator, modulating diverse cellular functions in the gut and peripheral tissues. Sodium butyrate markedly shapes the gut microenvironment by fortifying epithelial integrity, exerting anti-inflammatory effects—reducing pro-inflammatory cytokine production (e.g., IL-1β, IL-6, TNF-α), inhibiting NF-κB activation and modulating immune cell function—and by lowering luminal pH. Collectively, these mechanisms foster an ecosystem conducive to the growth and activity of beneficial bacteria, thereby supporting overall intestinal and immunological homeostasis [8].

There is a growing body of evidence demonstrating that inulin (a prebiotic) and sodium butyrate (a short-chain fatty acid) exert beneficial effects on gut health and can enhance the viability and functionality of probiotic strains. Inulin is a water-soluble oligosaccharide (a fructan) that can be extracted from various sources, including chicory, garlic, wheat, oats, and bulgur [9]. Structurally, it consists of β-(2→1)-linked fructosyl units terminating in a glucosyl residue [10]. As a recognized prebiotic, inulin is increasingly incorporated into fermented products such as yogurt to improve gastrointestinal health, enhance calcium absorption, and support immune function. Moreover, it stimulates the proliferation of probiotic genera—such as *Lactobacillus* and *Bifidobacterium*—during storage, thereby ensuring delivery of high numbers of viable cells to the colon [11]. Inulin has also been shown to improve the physicochemical, functional, and sensory properties of dairy matrices, while preserving probiotic viability in products like yogurt [12,13].

Sodium butyrate represents a key microbial metabolite with a broad spectrum of intestinal health–promoting activities. It fosters microbial diversity and supports the growth of beneficial bacteria while suppressing pathogenic species [14]. Although sodium butyrate is not directly utilized as a growth substrate by *Lactobacillus casei* or *Bifidobacterium*, it creates a favorable colonic environment by attenuating inflammation, reinforcing the intestinal barrier, and modulating luminal pH. Direct fortification of foods—including milk—with sodium butyrate is challenged by its characteristic odor; however, co-administration of inulin and sodium butyrate alongside probiotics in a fermented milk matrix yields a synbiotic formulation that may confer synergistic health benefits.

All probiotic strains used in this study are capable of fermenting milk as monocultures by converting lactose into lactic acid and lowering the pH. *Lactobacillus acidophilus* is well established as a traditional “acidophilus milk” culture, owing to its homofermentative metabolism that drives lactic acid production and significant pH reduction in milk-based systems [15]. Although *L. acidophilus* grows more slowly than standard yogurt starters, studies confirm its ability to acidify milk to pH values between 4.0 and 4.6 over extended fermentation periods [16]. *Lactobacillus casei* also possesses strong acidifying potential and has been widely used as a sole fermentative agent in dairy products such as fermented milks and probiotic drinks [17]. Its facultative heterofermentative metabolism allows for efficient lactic acid production under appropriate conditions, typically resulting in final pH values below 4.5 [18]. *Bifidobacterium animalis* subsp. *lactis*, while generally less acidifying than the aforementioned lactobacilli, also demonstrates the ability to ferment milk substrates. Pure cultures of *B. animalis* HN019 were able to reduce milk pH from 6.0 to approximately 5.3 within 14 h of fermentation [19]. Though slower to acidify, *B. animalis* contributes to pH reduction and metabolite production, especially when grown under anaerobic conditions or in combination with prebiotics.

Probiotic fermentations can yield substantial acidification of dairy substrates. For example, *Lacticaseibacillus casei* 431 rapidly ferments milk: in one study, a 2% inoculum of *L. casei* 431 drove pasteurized whole milk to pH 4.6 (from ~6.6), i.e., ΔpH 2.0, after 24 h at 37 °C [20]. Likewise, *Lactobacillus acidophilus* LA-5 is strongly acidifying: in a designed milk fermentation, it produced ΔpH 3.14 over 72 h at 37 °C (lowering pH to 3.6) [21]. *Bifidobacterium animalis* subsp. *lactis* BB-12 (when co-cultured with mesophilic dairy starters) showed comparable acidification, reaching pH 4.6 after incubation at 37 °C (ΔpH 2.0) [22]. Quantitative acidification data are scarcer for *Lacticaseibacillus paracasei* L26. In non-dairy tests, *L. paracasei* L26 lowered star fruit juice to pH 4.71 (starting from 5.2) after fermentation, indicating moderate lactic acid production [23].

Probiotic strains must overcome the hostile conditions of the human gastrointestinal tract—namely, gastric acidity (pH < 3.0) and bile salts (0.2–2.0% *w*/*v*)—in order to survive transit and exert health-promoting effects. Vernazza et al. [24] demonstrated that *Bifidobacterium animalis* ssp. *lactis* BB-12 suspended in glycine–HCl buffer (pH 2.0; 37 °C) retained >99% viability during the first 20 s, with only modest declines over longer exposures, whereas at pH 4.0 it maintained 81% viability after 20 min. In bile salt assays (0.5% Oxgall; 37 °C; 2 h), BB-12 reached 1.22 × 10^8^ CFU/mL versus 7.65 × 10^8^ CFU/mL in bile-free controls—equating to 16% survival—despite negligible changes in optical density, a phenomenon attributed to bile-induced cell autoaggregation [19]. Jacobsen et al. [25] showed that *Lactobacillus casei* 431^®^ retained > 10^5^ CFU/mL when exposed at pH 2.5 for 2 h followed by 0.3% Oxgall for 4 h (a <2 log_10_ reduction from 10^8^ CFU/mL). Salgaço et al. [26] found that *L. acidophilus* LA-5 suffered a 3.6 log_10_ CFU/mL reduction after 2 h in simulated gastric juice (0.3% HCl; pepsin 1 g/L; pH 2.0; 37 °C) and an additional 3.8 log_10_ loss after 4 h in simulated intestinal fluid (0.3% bile salts; pancreatin 1 g/L; pH 7.4; 37 °C), whereas microencapsulation in an ultra-filtered cheese matrix preserved viability at 6 log_10_ CFU/g through the full 6 h sequence [27] and planktonic cells tolerated 0.3% bile for 4 h with <1 log_10_ CFU/mL reduction [28]. In vitro assessment of *L. paracasei* L26 indicated a decline from 8.96 to 5.11 log_10_ CFU/mL (3.85 log drop) after 3 h at pH 2.0 (0.3% HCl; pepsin 1 g/L; 37 °C) and survival of 6.30 log_10_ CFU/mL at pH 3.0 (2.66 log drop), while exposure to 0.5% and 1.0% Oxgall for 6 h resulted in <1 log_10_ CFU loss from >10^8^ CFU/mL [29]. Despite its commercial use, no peer-reviewed data exist on the acid or bile tolerance of *Lactobacillus johnsonii* LJ. Available studies refer to other isolates (e.g., N5, N7, PF01), which survive pH 2.5 and 0.3% bile at rates of 33–114% over 2–4 h [30].

Recent advances have demonstrated that exogenously supplied short-chain fatty acids (SCFAs), particularly butyrate, can act synergistically with prebiotic fibers such as inulin to enhance epithelial barrier integrity and modulate gut immunity. Dietary carbohydrates that escape small-intestinal digestion are fermented by the colonic microbiota into SCFAs, which account for 60–150 mmol/kg luminal content in a healthy adult, with an acetate:propionate:butyrate ratio of ~60:25:15. These metabolites serve not only as major energy substrates for colonocytes but also as signaling molecules that regulate tight-junction protein expression, mucin secretion, and anti-inflammatory pathways [31,32,33]. Despite growing interest in synbiotic formulations, there remains a paucity of studies systematically evaluating the co-fortification of fermented dairy matrices with both inulin and sodium butyrate across multiple clinically relevant probiotic strains. Inulin acts as a selective substrate for bifidobacteria and lactobacilli, enhancing their storage viability and bile tolerance. Sodium butyrate, the sodium salt of butyric acid, further supports mucosal homeostasis and may potentiate probiotic colonization by lowering luminal pH and inducing antimicrobial peptide production [32,33].

Therefore, this work aims to fill that gap by evaluating the physicochemical, sensory, and microbiological outcomes of milk fermentation when co-supplemented with 4% (*w*/*w*) inulin and 0.06% (*w*/*w*) sodium butyrate across five clinically relevant probiotic strains. We further assess the survival of these strains under a standardized static in vitro digestion model simulating oral, gastric, and intestinal phases, providing novel insights for the design of functional dairy supplements.

In vitro assessment of probiotic viability in sodium butyrate–enriched milk matrices is expected to yield critical insights into their therapeutic potential for dietary interventions and microbiota restoration.

## 2. Materials and Methods

### 2.1. Materials

Milk used in the experiments (Łaciate, SM Mlekpol, Grajewo, Poland) had the following composition per 100 mL: fat 2.0 g, lactose 4.8 g, protein: 3.3 g, salt 0.10 g, calcium: 120 mg. Three formulations were prepared: control (no supplementation), inulin-fortified (4%, *w*/*w*), and inulin + sodium butyrate–fortified (4% and 0.06%, *w*/*w*, respectively). Inulin was procured from TAR-GROCH-FIL (Filipowice, Poland), and sodium butyrate was obtained from OstroVit Sp. z o.o. (Zambrów, Poland). The probiotic strains employed included *Bifidobacterium animalis* subsp. *lactis* BB-12, *Lacticaseibacillus casei* 431, *Lacticaseibacillus paracasei* L26, *Lactobacillus acidophilus* LA-5 (Chr. Hansen, Hoersholm, Denmark), and *Lactobacillus johnsonii* LJ Delvo^®^Pro (DSM, Delft, The Netherlands). Growth media and reagents—such as MRS agar, peptone water, NaOH, and phenolphthalein—were supplied by Biocorp (Warsaw, Poland) and Chempur (Piekary Śląskie, Poland). Enzymes and chemicals for the in vitro digestion protocol (α-amylase, mucin, pepsin, bile extract, and pancreatin) were purchased from Sigma-Aldrich (St. Louis, MO, USA), while all analytical-grade salts and acids were provided by Chempur (Piekary Śląskie, Poland).

### 2.2. Sample Preparation

Each milk batch (control, inulin-fortified, and inulin + sodium butyrate–fortified) was homogenized at 60 °C under 20 MPa, then re-pasteurized at 85 °C for 10 min. After cooling to 37 ± 1 °C, the milk (Łaciate, SM Mlekpol, Grajewo, Poland) was inoculated (5%, *w*/*w*) with a single, active probiotic strain (activated to ~log 9 CFU g^−1^)—either *B. animalis* BB-12, *L. johnsonii* LJ, *L. casei* 431, *L. paracasei* L26, or *L. acidophilus* LA-5—following a 5 h activation step at 40 °C. Inoculated milk samples were gently mixed, aliquoted into 100 mL polypropylene containers, and incubated at 37 °C until they reached a pH of 4.60 ± 0.10. After fermentation, all samples were promptly chilled to 5 °C in a refrigerated incubator (Cooled Incubator ILW 115, POL-EKO Aparatura, Wodzisław Śląski, Poland). For each experimental condition (Table 1), a 2000 mL batch of inoculated milk (control, inulin-fortified, and inulin + sodium butyrate–fortified) was dispensed into twenty 100 mL polypropylene containers (n = 20) to provide sufficient material for all analyses and in vitro digestion. Following 24 h of storage at 5 °C, samples were subjected to physicochemical and sensory evaluations, as well as in vitro digestion and microbiological analyses.

### 2.3. Acidity

pH was measured with a FiveEasy™ pH meter (Mettler-Toledo, Greifensee, Switzerland) equipped with InLab^®^ Solids Pro-ISM electrodes (Mettler-Toledo, Greifensee, Switzerland), directly in 100 mL of fermented milk.

Titratable acidity was determined following ADPI Method #007a [34], with modification. Briefly, 25 g of fermented milk was titrated with 0.1 N NaOH in the presence of 0.5 mL of phenolphthalein until the first persistent (30 s) faint pink color was achieved. TA (% lactic acid *w*/*v*) was calculated as$$\mathrm{TA}\;(\%\;\mathrm{lactic}\;\mathrm{acid})=\frac{NaOH\;titrant,\;mL\times normality\;of\;NaOH\times 9.008}{sample\;weight,\;g}$$

### 2.4. Syneresis

Syneresis was evaluated by quantifying the volume of whey expelled relative to the initial sample mass. Briefly, a 10 g portion of fermented milk was transferred into a 50 mL polypropylene tube and centrifuged at 3160× *g* for 10 min at 5 °C (Refrigerated Centrifuge LMC-4200R; Biosan SIA, Rīga, Latvia) [35]. The amount of separated whey was then weighed and reported as a percentage of the original sample weight.

### 2.5. Color

Instrumental color measurements of fermented milk were performed using a Precision Colorimeter (Model NR 145, Shenzhen, China), following the CIELab color-space methodology. Prior to analysis, the device was calibrated using a standardized white tile [36]. Lightness (L*) was recorded on a scale from 0 (black) to 100 (white). The a* coordinate (negative values indicating green, positive indicating red) and the b* coordinate (negative values indicating blue, positive indicating yellow) were also recorded. From these primary coordinates, chroma (C*)—reflecting color saturation—and hue angle (h°) were calculated to describe color purity and hue position, respectively. For each measurement, 100 mL of fermented milk was placed in a plastic cup on a white background, and the colorimeter probe was positioned directly on the sample surface; the instrument then displayed L*, a*, b*, C*, and h° values.

### 2.6. Organoleptic Evaluation

The organoleptic evaluation was conducted in a dedicated sensory-analysis laboratory by a trained panel of 30 individuals (17 women and 13 men, aged 23–55). Each panelist was seated in an individual sensory booth and provided with a set of three randomly three-digit-coded fermented milk samples and an evaluation form. Panelists tasted and/or smelled each sample in sequence and recorded their perceptions by selecting the corresponding point on a nine-point scale with anchored endpoints (1 = least intense/least characteristic; 9 = most intense/most characteristic). The following attributes were evaluated: consistency, milky-creamy taste, sour taste, sweet taste, off-taste, fermentation odor, sour odor, and off-odor [37,38]. Definitions of attributes in the descriptive organoleptic analysis of fermented milk [39]: 

Milky-creamy taste: a taste evoked by milk powder.

Sour taste: taste evoked by lactic acid.

Sweet taste: taste evoked by sucrose.

Off-taste: an atypical or uncharacteristic taste.

Fermentation odor: intensity of odor associated with sour milk and freshly fermented dairy products.

Sour odor: odor evoked by acids.

Off-odor: an atypical or uncharacteristic odor.

### 2.7. In Vitro Digestion Protocol

A static in vitro gastrointestinal digestion model (oral, gastric, and intestinal phases) was employed based on established protocols [40,41], with minor modifications. For each formulation of fermented milk, three digestion experiments were conducted in parallel, each using an independent 50 mL aliquot. Each aliquot was processed up to a defined phase—oral, gastric, or intestinal—and terminated accordingly for subsequent analysis. All digestion phases were performed in triplicate per formulation to ensure reproducibility.

During the oral phase, each 50 mL aliquot of fermented milk was combined with 5 mL of simulated salivary fluid in a 100 mL amber glass vessel. The simulated salivary fluid was prepared by dissolving 2.38 g Na_2_HPO_4_, 0.19 g K_2_HPO_4_, 8.0 g NaCl, 100 mg mucin, and 150 mg α-amylase (200 U/L activity) in 1 L of deionized water. The mixtures were adjusted to pH 6.75 ± 0.20 with 1 M NaOH and incubated at 37 °C for 10 min with constant stirring (90 rpm) to simulate oral digestion.

Gastric digestion was then initiated by adding pepsin (13.08 mg per 50 mL aliquot) to each oral bolus and acidifying the mixture to pH 2.00 ± 0.20 with 12 M HCl. The samples were incubated at 37 °C for 2 h under constant stirring (90 rpm) to simulate gastric conditions.

Finally, to simulate the intestinal phase, 5 mL of pancreatin solution (4 g/L in deionized water) and 5 mL of bile salt solution (25 g/L) were added to each gastric digest. The pH was raised to 7.00 ± 0.20 (using 1 M NaOH), and incubation was continued at 37 °C for an additional 2 h with constant stirring (90 rpm).

Each digestion was halted at its designated end point (oral, gastric, or intestinal phase), and the resulting digesta were collected immediately for microbiological analysis as described in Section 2.8.

### 2.8. Microbiological Analysis

Probiotic viability was determined both prior to and following each simulated digestive stage (oral, gastric, and intestinal). For enumeration, 10 g of each fermented milk sample was homogenized in 90 mL of sterile 0.1% peptone water (10^−^^1^ initial dilution), followed by serial dilution (adjusted to the experimental stage). Appropriate dilutions were pour-plated onto MRS agar and incubated under anaerobic conditions at 37 °C for 72 h using a GENbox anaerobic system (Biomerieux, Warsaw, Poland) within an incubator. After incubation, colonies were enumerated with a TYPE J-3 colony counter (Chemland, Stargard Szczeciński, Poland) and expressed as log CFU g^−^^1^. Probiotic survival rate (%) was calculated according to the following equation [41]:$$\mathrm{Survival}\;\mathrm{rate}\;(\%)=\frac{Viable\;counts\;in\;digested\;sample}{Viable\;counts\;in\;undigested\;sample}\;\times 100$$

### 2.9. Statistical Analysis

Data processing was carried out in Statistica 13.1 (StatSoft, Tulsa, OK, USA). Results are expressed as mean and standard deviation (SD). To determine significant differences, Tukey’s test was applied at a threshold of *p* ≤ 0.05. In addition, one-way and multifactorial analyses of variance (ANOVA) were performed, along with correlation analyses, to assess the effects of treatments and interrelationships between variables. Each experimental condition was replicated three times, with five technical replicates per experiment.

## 3. Results and Discussion

Worldwide, an increasing variety of probiotic-enriched foods (and their metabolites) are both produced and consumed [42]. Among these, dairy products remain the most prevalent delivery vehicles for probiotics, owing to the favorable adaptation of such microorganisms within the milk matrix [43] and their broad consumer acceptance [44,45]. Although sodium butyrate naturally occurs in dairy items, its deliberate addition to milk prior to fermentation alters the product’s microenvironment and can influence its physicochemical and sensory attributes. Consequently, we elected to compare unmodified milk to samples fortified with inulin or with a combined inulin–sodium butyrate combination.

The colorimetric outcomes for fermented milk are summarized in Table 2. Lightness (L*) values ranged from 90.72 (for the PI formulation) to 98.45 (for CA). In our experiments, milk fermented by *Bifidobacterium animalis*, *Lactobacillus johnsonii*, and *Lactobacillus acidophilus* exhibited significantly greater L* values than samples fermented by *Lacticaseibacillus casei* and *Lacticaseibacillus paracasei*. Both inulin and the inulin–butyrate combination induced a perceptible darkening of the fermented milk samples compared to their respective controls, regardless of the probiotic strain employed.

González-Tomás et al. [46] previously reported that the average chain length of added inulin predominantly affected lightness (L*), without significantly altering any other color coordinates. Specifically, their long-chain inulin-supplemented and unsupplemented samples were the brightest in that study.

In this study, all fermented milk samples—both control and supplemented—displayed negative a* values, indicating a stronger green component relative to red. The most pronounced greenness was observed in the BbI sample (a* = −3.10). Notably, the addition of either inulin or the inulin–butyrate combination led to a significant intensification of green hue only in the *L. acidophilus* groups.

The yellow–blue coordinate (b*) was dependent on the probiotic strain. The highest yellow chromaticity (b* > 10) occurred in milk samples fermented by *B. animalis*, *L. johnsonii*, and *L. acidophilus*. Conversely, *L. casei* and *L. paracasei* samples yielded significantly lower b* values. Inulin or the combined inulin-butyrate supplementation caused a statistically significant decrease in yellow intensity exclusively in the *B. animalis* and *L. acidophilus* fermented milk samples.

Finally, chroma (C*) values ranged from 7.88 (CI) to 11.57 (CBb), while hue angle (h°) varied from 99.47 (CA) to 106.05 (JI).

The strain-dependent differences in color parameters observed in our study can be attributed to variations in fermentation metabolism, particularly in acid production and its effects on the milk matrix. Statistical analysis (Table 2) confirmed that both strain type and supplementation significantly influenced (*p* < 0.05) the color parameters L* and b*. Specifically, fermented milks inoculated with *B. animalis*, *L. johnsonii*, and *L. acidophilus*—which produced higher terminal pH values (4.5)—exhibited significantly greater lightness (L*) and yellow hue (b*) compared to those fermented by more strongly acidifying strains such as *L. casei* and *L. paracasei*, which reached pH values of 4.2–4.3. These latter samples appeared darker (lower L*) and less yellow (lower b*).

This pattern aligns with established relationships between pH, casein aggregation, and visual appearance. As the pH of fermenting milk approaches the isoelectric point of casein (4.6), protein precipitation and whey separation (syneresis) intensify, reducing the brightness and increasing the translucency of the fermented matrix [47,48]. Conversely, a slightly higher pH results in a looser curd structure and greater whey retention, leading to a lighter, more opaque appearance.

In addition, we hypothesize that differences in exopolysaccharide (EPS) production, although not directly measured in this study, may have contributed to the observed color variation. Strains such as *L. acidophilus* and *B. animalis* are known to produce EPS, which enhance water-holding capacity and increase light scattering by creating a porous protein matrix [48]. Conversely, strains with lower EPS production—such as *L. casei* and *L. paracasei*—likely formed denser curds with more whey separation, resulting in a darker and less turbid appearance. According to Cais Sokolińska and Pikul [49], the microbiological activity of yogurt microflora leads to changes in the potential acidity of the clot, and thus causes physical changes in the clot, resulting in a difference in color brightness and the degree of color saturation. These physiological and metabolic traits, together with acidification profiles, explain the statistically significant differences in color parameters observed among probiotic-fermented milk samples.

Table 3 presents the pH values. All milk samples supplemented with inulin exhibited significantly lower pH compared to their respective controls, independent of the probiotic strain applied. The lowest pH (4.23) was observed in the *L. paracasei* culture with inulin (PI), whereas the highest pH (4.63) was recorded in the control *B. animalis* sample (CBb). Inclusion of the inulin-sodium butyrate combination led to a significant pH reduction only in the milk fermented by *B. animalis* and *L. acidophilus*. Three-way ANOVA (Table 6) confirmed that pH was strongly influenced by all main factors (probiotic strain, inulin supplementation, and inulin–butyrate supplementation) and their interactions.

Consistent with our results, Cruz et al. [50] reported pH values of 4.1–4.2 in yogurts supplemented with inulin.

Lactic acid concentration in fermented milk depended on the fermenting strain (Table 3). In samples CP, PIM, CC, CIM, JI, JIM, CA, and AI, lactic acid levels were approximately 0.9%. In contrast, milk fermented by *B. animalis*—both control and supplemented—showed the lowest lactic acid content, particularly in sample CBb. In Helal et al. [51] study, yogurt titratable acidity (TA%) measured post-fermentation ranged from 0.72 to 0.82%. No significant difference in TA% was detected between low-fat and whole-milk yogurt formulations, and inulin supplementation exerted no observable effect on acidity. The TA% values reported by Helal et al. [51] correspond closely to those obtained in the present study.

Numerous studies have examined how prebiotics affect the chemical composition of dairy products. Inulin’s pronounced acidifying effect can be mechanistically attributed to its role as an additional fermentable substrate for lactic acid bacteria. Several authors have shown that *Lactobacillus* spp. possess inulinase and β-fructofuranosidase activities, enabling them to hydrolyze inulin into fructose units which are subsequently converted to l- and d-lactic acid via glycolysis, thus accelerating pH decline to values near 4.0–4.2 [52]. Hardi and Slacanac [53] reported that incorporation of inulin (4–6% *w*/*w*) into milk markedly hastened pH drop during fermentation, corroborating our observation of significantly lower end-point pH in all inulin-supplemented samples regardless of strain. Strain-dependent utilization of inulin explains the differential effects observed among probiotic cultures. *B. animalis*, which channels oligosaccharides primarily toward acetic and formic acid production, exhibits lower lactic acid yield and thus a higher residual pH compared to *L. paracasei* or *L. acidophilus* [54]. This metabolic divergence is consistent with our finding that the lowest pH (4.23) occurred in the *L. paracasei*  +  inulin group (PI), whereas the highest pH (4.63) was measured in the *B. animalis* control (CBb). Nevertheless, not all studies report an inulin-induced increase in acidity: Helal et al. [51] and Guven et al. [55] found no significant impact of inulin on titratable acidity or pH in low-fat and whole-milk yogurts, Wszołek [56] similarly observed unchanged pH in bio-yogurts fortified with inulin, and Nastaj and Gustaw [57] even reported that supplementing with inulin and oligofructose prolonged the acidification time during the production of set-style yogurts. Such discrepancies likely reflect differences in inulin degree of polymerization, concentration (typically 2–8% *w*/*w*), fermentation temperature, and matrix composition (fat content, solids non-fat), which can modulate prebiotic accessibility and bacterial metabolism. Taken together, these data support the notion that inulin supplementation intensifies lactic acid production—and hence pH reduction—in a manner that is both strain- and formulation-dependent.

Sodium butyrate at 0.06% *w*/*w* dissociates in the milk matrix to release H^+^ ions, directly contributing to the acid pool and thereby lowering pH and increasing titratable acidity [58]. Moreover, supplementation of dairy fermentations with butyrate salts has been shown to modulate lactic acid bacterial metabolism—enhancing organic-acid production rates and accelerating overall acidification kinetics [59].

Syneresis (a common occurrence in fermented dairy products like yogurt) is an important quality indicator for acid-set systems [60]. Dmytrów et al. [61] noted that the choice of starter culture markedly influences syneresis in acid-coagulated milk. In this study, both probiotic strain and added supplements modulated syneresis in fermented milk. As shown in Table 3, inulin and the combined inulin–butyrate supplementation significantly reduced syneresis in milk fermented by *L. johnsonii*, *L. acidophilus*, *L. casei*, and *L. paracasei* compared to their control counterparts. No such reduction was observed for *B. animalis* samples. Control milk fermented by *L. casei* and its supplemented variants (CC, CI, CIM) exhibited the lowest syneresis levels, whereas the highest syneresis was found in the control *L. johnsonii* sample (CJ) and in all *B. animalis* groups (CBb, BbI, BbIM).

According to Ipsen et al. [62], an optimal inulin concentration can mitigate whey separation. Other investigations have demonstrated that inulin reduces syneresis in low-fat and fat-free yogurts [63,64]. Likewise, Cruz et al. [50] reported that inulin supplementation in fermented dairy products lowers syneresis while enhancing product cohesiveness and firmness. In contrast, Bisar et al. [65] argued that inulin may actually exacerbate syneresis in yogurt due to its chemical composition—characterized by a higher fructose-to-glucose unit ratio and low degree of branching—which weakens protein–carbohydrate interactions. Furthermore, syneresis has been linked to yogurt porosity [66], and increased porosity can, in turn, promote greater whey expulsion [67]. Although specific studies on sodium butyrate’s effect on syneresis in fermented milk are scarce, analogous work on small organic acids—e.g., conjugated linoleic acid—demonstrates that these molecules can interact with casein micelles, promoting protein cross-linking and a more compact gel microstructure, thereby reducing whey separation [68]. By extension, sodium butyrate may similarly bind to milk proteins and enhance network stability, which—together with inulin’s water-binding properties—could account for the markedly lower syneresis observed in our inulin–butyrate treatments.

Tungland and Meyer [69] have reported that inulin also offers valuable technological functionality, serving as a structuring ingredient within food matrices.

The sensory assessment (Table 4) demonstrated that, for most probiotic strains, control samples (CP, CC, CBb, CA) showed better textural properties compared to their supplemented counterparts—except in the case of CJ. Specifically, the *L. johnsonii* sample with inulin–butyrate (JIM) scored higher for consistency than its control (CJ). Moreover, for milk fermented by *B. animalis*, both inulin and the inulin–butyrate combination significantly reduced consistency ratings.

Inulin fortification of yogurt imparts several sensory advantages, notably a richer, cream-fat perception in low-fat formulations [70]. Moreover, inulin has been shown to modulate the yogurt’s flavor profile during storage—reducing acetaldehyde levels while elevating volatile fatty acid concentrations—via the enzymatic activity of the yogurt starter culture [55].

Gonzalez-Tomas et al. [46] found that long-chain inulin increased perceived mouthfeel roughness in inulin-enriched dairy desserts; they attributed this effect to small crystals or crystalline aggregates of long-chain inulin affecting sample texture. Similarly, Villegas and Costell [71] observed that both the chain length and concentration of inulin influenced the rheological behavior of milk-based beverages, with outcomes depending on whether skimmed or whole milk was used, as well as on κ-carrageenan addition. Their results showed that skimmed-milk beverages containing 4–10% short-chain inulin, 6–8% native inulin, or 4–6% long-chain inulin achieved viscosities comparable to whole-milk counterparts. Other researchers have examined how varying inulin levels affect rheological and sensory characteristics in products such as yogurts [63] and milk-based desserts [72,73]. Accordingly, the sandy mouthfeel reported by our panelists may stem from both the type of inulin employed and the quantity added.

In our study, supplementation with inulin or the inulin-butyrate mixture significantly diminished the milky-creamy taste compared to controls only in milks fermented by *L. casei* and *L. paracasei*. Sour taste ratings ranged from 4.25 for CP and JI to 6.00 for CIM and AIM. In these cases, the addition of inulin–butyrate notably heightened sour taste intensity while reducing perceived sweetness. All control samples exhibited a sweeter profile than their supplemented counterparts, although sweet taste intensity varied by strain—from 6.25 for CJ to 4.40 for CA and CC. Interestingly, milks supplemented with inulin-butyrate (PIM, CIM, BbIM, JIM, AIM) consistently demonstrated a stronger sour odor compared to their respective controls. No off-tastes or off-odors were detected in any samples.

Table 5 and Table 6 summarize the viability (log CFU g^−1^) of five probiotic strains in milk under varying initial cell densities, inulin, and inulin–butyrate supplementation. Maintaining a high initial cell count (commonly >10 log CFU g^−1^) is essential to ensure that the final probiotic dose exceeds the therapeutic threshold (>10^6^ CFU g^−1^) after processing and storage [74,75]. Statistically significant improvements in survival conferred by inulin or butyrate support the development of standardized synbiotic formulations with consistent efficacy [76]. Specifically, inulin acts as a fermentable substrate that enhances bacterial resilience during storage and gastrointestinal transit [5], while butyrate contributes to gut barrier integrity and exerts anti-inflammatory effects in the host epithelium [77]. Moreover, our three-factor ANOVA not only confirms the main effects of initial inoculum level, inulin, and butyrate, but also reveals synergistic interactions among these variables, providing a rational framework for optimizing probiotic milk products tailored to each strain [75].

To evaluate the impact of inulin and the inulin-sodium butyrate combination on the viability of *Lacticaseibacillus paracasei*, *Lacticaseibacillus casei*, *Bifidobacterium animalis*, *Lactobacillus johnsonii*, and *Lactobacillus acidophilus* during three-stage in vitro digestion, bacterial counts were determined in each milk sample prior to digestion. Detailed enumeration results at each simulated digestive phase are provided in Table 5.

**Table 6 nutrients-17-02249-t006:** Analysis of variance (ANOVA) *p*-value on effect of probiotic strains, inulin, inulin and sodium butyrate on color (L*, a*, b*, C*, h°), pH, lactic acid, syneresis, organoleptic parameters (consistency, milky-creamy taste, sour taste, sweet taste, off-taste, sour odor, off-odor), and viable counts of bacteria in milk fermented by probiotic bacteria before digestion, in the oral cavity, stomach, and small intestine.

		Probiotic Strains *p*-Values	Inulin *p*-Values	Inulin and Sodium Butyrate *p*-Values	Probiotic Strains* Inulin *p*-Values	Probiotic Strains*Inulin and Sodium Butyrate *p*-Values	Inulin*Inulin and Sodium Butyrate *p*-Values	Probiotic Strains*Inulin* Inulin and Sodium Butyrate *p*-Values
L*		0.0000↑	0.0000↑	0.0000↑	0.1895 n.s.	0.0076↑	0.0546 n.s.	0.0141↑
a*		0.0000↑	0.0000↑	0.0147↑	0.0083↑	0.0196↑	0.0492↑	0.0012↑
b*		0.0000↑	0.0029↑	0.4285 n.s.	0.0000↑	0.0039↑	0.3121 n.s.	0.1281 n.s.
C*		0.0000↑	0.9633 n.s.	0.5738 n.s.	0.0766 n.s.	0.0176↑	0.0512 n.s.	0.4362 n.s.
h°		0.0000↑	0.0384↑	0.0001↑	0.0000↑	0.0000↑	0.0011↑	0.0123↑
pH		0.0000↑	0.0000↑	0.0002↑	0.0000↑	0.0003↑	0.0021↑	0.0041↑
TA %		0.0000↑	0.2531 n.s.	0.0000↑	0.1118 n.s.	0.0000↑	0.0002↑	0.0518 n.s.
Syneresis		0.0000↑	0.0001↑	0.0304↑	0.0007↑	0.2215 n.s.	0.0041↑	0.0412↑
Consistency		0.0000↑	0.0524 n.s.	0.0501 n.s.	0.4901 n.s.	0.2111 n.s.	0.8529 n.s.	0.1653 n.s.
Milky-creamy taste		0.0025↑	0.1421 n.s.	0.0868 n.s.	0.5535 n.s.	0.2405 n.s.	0.5700 n.s.	0.2032 n.s.
Sour taste		0.5283	0.1487 n.s.	0.0491↑	0.1739 n.s.	0.02170↑	0.4669 n.s.	0.8357 n.s
Sweet taste		0.0000↑	0.1232 n.s.	0.0432↑	0.1323 n.s.	0.0510 n.s.	0.7233 n.s.	0.5349 n.s.
Off-taste		0.1210 n.s.	0.0539 n.s.	0.0511 n.s.	0.6929 n.s.	0.9231 n.s.	0.3055 n.s.	0.5797 n.s.
Sour odor		0.0166↑	0.8735 n.s.	0.0001↑	0.0530 n.s.	0.0490↑	0.0623 n.s.	0.1818 n.s.
Off-odor		0.3047 n.s.	0.7681 n.s.	0.8153 n.s.	0.2370 n.s.	0.8460 n.s.	0.3317 n.s.	0.7190 n.s.
Viable counts of bacteria in milk fermented by probiotic bacteria	Before digestion	0.0000↑	0.0182↑	0.0000↑	0.0058↑	0.0000↑	0.0211↑	0.0011↑
	Oral cavity	0.0000↑	0.0000↑	0.2060↑	0.0000↑	0.0070↑	0.0034↑	0.0003↑
	Stomach	0.0000↑	0.0485↑	0.0179↑	0.0103↑	0.0003↑	0.0211↑	0.0411↑
	Small intestine	0.0000↑	0.0002↑	0.0408 ↑.	0.0064↑	0.0063 ↑.	0.0044 ↑	0. 0101 ↑

Probiotic Strains*Inulin = interaction ↑; Probiotic Strains*Inulin and Sodium butyrate= interaction ↑; Inulin*Inulin and Sodium butyrate = interaction ↑; Probiotic Strains*Inulin*Inulin and Sodium butyrate = interaction ↑; *p* < 0.05 indicates significant effect; n.s.—no significant effect.

Initial viable counts ranged from 10.42 log CFU g^−1^ (PI) to 11.97 log CFU g^−1^ (CJ). Before digestion, the highest populations were observed in milk fermented by *L. casei* and *L. johnsonii*. For these two strains, adding either inulin or inulin and sodium butyrate significantly resulted in a significant decrease in their population density. A three-way ANOVA (Table 6) confirmed that all main factors—bacterial strain, inulin supplementation, and inulin-butyrate supplementation—as well as their interactions, exerted a significant effect on viable counts in the milk.

In the oral phase, probiotic cells must withstand not only mechanical shear forces from mastication but also the action of salivary α-amylase, which—while not directly bactericidal—can cleave extracellular polysaccharides and thereby modulate bacterial adhesion to the mucosal surface [78]. During the oral phase, as anticipated, no statistically significant reduction in probiotic counts was detected; values remained above 10 log CFU g^−1^ for all formulations. Likewise, the influence of each experimental factor on probiotic levels persisted at this stage (Table 6).

Because gastric digestion typically lasts 1–3 h depending on the food matrix, a 2 h incubation in simulated gastric juice was used. Probiotic bacteria must also demonstrate resilience against host defense mechanisms—acidic pH, bile salts, and digestive enzymes [79]. As shown in Table 5, gastric survival varied considerably among strains. *L. casei* proved least acid-resistant: the CC sample retained only 1.43 log CFU g^−1^ at the end of the gastric phase, indicating a 9.63 log CFU g^−1^ reduction relative to pre-digestion counts. However, static in vitro gastric models maintain constant pH and enzyme concentrations, whereas the human stomach exhibits dynamic peristaltic contractions and intermittent chyme influx that can attenuate localized acid stress, thereby mitigating probiotic cell damage [80,81]. Moreover, prebiotic inulin can form a transient hydrophilic polysaccharide network around bacterial cells, impeding protease access and enhancing mucoadhesion to the gastric mucosa, which contributes to improved survival under proteolytic conditions [82]. In contrast, inclusion of inulin or the inulin–butyrate combination in *L. casei* milk (CI and CIM) improved its gastric survival. The most robust strains under acidic challenge were *B. animalis* and *L. johnsonii*, both maintaining counts above 6 log CFU g^−1^ during the gastric phase. Supplementation effects varied by strain: inulin significantly enhanced gastric survival of *B. animalis* and *L. paracasei*, while the inulin–butyrate mixture provided a beneficial effect for *L. acidophilus*, *B. animalis*, and *L. johnsonii*. ANOVA confirmed that strain, inulin, and inulin–butyrate, and their interactions, had significant impacts on survival during the gastric stage.

The pronounced reduction in viable counts during the simulated gastric phase can be attributed to the synergistic lethality of an extremely low pH (≈2) and the proteolytic activity of pepsin, which compromise bacterial envelope integrity and denature essential enzymes, causing many cells to become non-culturable or die outright [83]. A significant fraction of the population, however, survives as sub-lethally injured cells: although unable to form colonies under acute acid stress, these cells retain the capacity for repair and resuscitation once they encounter milder conditions. Upon transition to the small-intestinal simulation (where pH is neutralized to 7.0 and pancreatin plus bile salts exert milder, more selective pressure), these injured cells can restore membrane integrity, reactivate stress-response systems (e.g., chaperone proteins and proton-pumping F_0_F_1_-ATPases), and resume cell division, leading to an apparent “rebound” in CFU [83,84]. Moreover, many *Lactobacillus* and *Bifidobacterium* strains express bile salt hydrolases (BSHs) that deconjugate and detoxify bile acids, further enhancing survival and modest growth in the intestinal milieu [85]. Finally, the presence of prebiotic substrates such as inulin provides fermentable carbon sources and may exert an osmoprotective effect, facilitating bacterial recovery and limited proliferation during the two-hour intestinal incubation [86].

Talearngkul et al. [87] likewise reported a 1 log CFU g^−1^ reduction for *L. acidophilus* LA-5 and *B. animalis* BB-12 in simulated gastric conditions. Acid stress can damage bacterial cell membranes, DNA, and proteins; thus, acid resistance remains a key probiotic selection criterion [45].

In the ensuing intestinal phase, bacteria encounter bile-containing intestinal fluid (approximately 0.7 L/day; pH ~8.0; mineral salt concentration ~0.5%). Bile salts and digestive enzymes primarily dictate survival by disrupting cell membranes [88]. In our study, only the BbIM, JIM, and AIM samples (those supplemented with inulin–butyrate) did not show a significant population increase compared to the gastric phase; all other formulations exhibited a statistically significant rise in counts, with all samples exceeding 5 log CFU g^−1^ by the end of the intestinal stage. For example, inulin fermentation by bifidobacteria yields a characteristic mixture of acetate, propionate, and butyrate that lowers the luminal pH, suppressing opportunistic pathogens and supporting mucosal health [5,89]. Moreover, butyrate—being the preferred energy substrate of colonocytes—upregulates tight-junction proteins and mucin production, thereby reinforcing epithelial barrier integrity and anti-inflammatory homeostasis in the distal gut [90].

Ziarno and Zaręba [86] demonstrated that inulin, in nearly every tested dose, stimulated bifidobacterial cells to develop greater environmental tolerance. Certain prebiotics—such as inulin—exert protective effects on bacterial cells, enhancing probiotic survival in the colon, especially for bifidobacteria.

Our findings align with previous reports showing strain-dependent survival differences under in vitro gastrointestinal simulation (Figure 1). Some strains exhibit high resilience, whereas others are more susceptible. In this study, *L. acidophilus*, *B. animalis*, and *L. johnsonii* displayed the most favorable survival percentages (all above 55%) compared to their respective pre-digestion counts. Inulin significantly improved the survival of *L. paracasei*, *B. animalis*, and *L. acidophilus* by 13.5%, 7.0%, and 13.0%, respectively, relative to controls. The inulin–butyrate supplementation specifically enhanced the survival of *L. casei* and *L. acidophilus* by 8.2% and 6.8%. Neither additive significantly affected *L. johnsonii*, which maintained survival above 61%. Again, ANOVA (Table 6) revealed that strain, inulin, and inulin-butyrate—and their interactions—significantly influenced overall probiotic survival under simulated digestion.

Szopa et al. [91] found that *L. casei* and *L. paracasei* exhibited the highest survival (>50%) under similar in vitro gastrointestinal conditions, whereas *L. rhamnosus* showed the greatest population decline, likely due to low exopolysaccharide (EPS) production, which otherwise confers protection against acid and bile [92,93]. Radicioni et al. [94] also suggested that differences in cell-surface composition and EPS biosynthesis capacity may explain the superior survival of *L. paracasei* and *L. casei*.

Initial inoculum density likewise influences survival outcomes [95,96]: larger starting populations yield greater numbers of cells capable of withstanding intestinal conditions. Additionally, strain selection remains critical for GIT survival [97,98,99]. Vernazza et al. [24] illustrated that among five *Bifidobacterium* strains, *B. animalis* subsp. *lactis* BB-12 displayed the highest resistance to low pH and bile salts.

## 4. Conclusions

The present study demonstrates that probiotic-fermented cow’s milk constitutes an effective vehicle for co-delivery of prebiotics and sodium butyrate. The impact of inulin-butyrate fortification on product characteristics was strain-dependent (namely *Bifidobacterium animalis* subsp. *lactis* BB-12, *Lacticaseibacillus casei* 431, *Lacticaseibacillus paracasei* L26, *Lactobacillus acidophilus* LA-5, and *Lactobacillus johnsonii* LJ).

Both inulin and the inulin–butyrate combination induced a perceptible darkening of the fermented milk samples compared to their respective controls, regardless of the probiotic strain employed. Notably, the addition of either inulin or the inulin–butyrate combination led to a significant intensification of green hue only in the *L. acidophilus* groups. Inclusion of the inulin-sodium butyrate combination led to a significant pH reduction only in the milk fermented by *B. animalis* and *L. acidophilus*. Inulin and the combined inulin–butyrate supplementation significantly reduced syneresis in milk fermented by *L. johnsonii*, *L. acidophilus*, *L. casei*, and *L. paracasei* compared to their control counterparts. In an organoleptic evaluation, no off-taste or off-odor was detected in any sample, even despite the addition of sodium butyrate.

Importantly, incorporation of this prebiotic–butyrate combination did not compromise probiotic viability during in vitro gastrointestinal simulation. These findings offer valuable guidance for the formulation of functional dairy supplements and fortified foods, enabling the design of products with enhanced health benefits tailored to support the gut microbial ecosystem.

Collectively, our data demonstrate that inulin and inulin-sodium butyrate supplementation markedly improve the gastrointestinal survival of all five tested probiotic strains in a strain-dependent fashion, as revealed by three-factor ANOVA. Inulin serves dually as an osmoprotectant and fermentable substrate, while sodium butyrate enhances barrier-related functions in the gut, together mitigating acid and enzymatic stresses. These insights pave the way for synbiotic milk formulations with finely tuned inoculum densities and additive levels to ensure reliable probiotic viability and functional performance.

## Figures and Tables

**Figure 1 nutrients-17-02249-f001:**
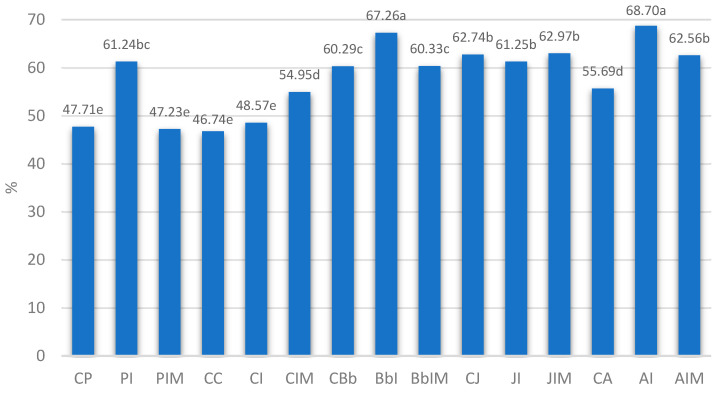
Survival rates (%) in milk fermented by probiotic bacteria. Mean ± standard deviation; ^a–e^—mean values denoted by different letters differ statistically significantly at *p* ≤ 0.05. **CP**—control milk fermented by *L. paracasei*; **PI**—milk with 4% inulin fermented by *L. paracasei*, **PIM**—milk with 4% inulin and 0.06% sodium butyrate fermented by *L. paracasei*, **CC**—control milk fermented by *L. casei*, **CI**—milk with 4% inulin fermented by *L. casei*, **CIM**—milk with 4% inulin and 0.06% sodium butyrate fermented by *L. casei*, **CBb**—control milk fermented by *B. animalis*, **BbI**—milk with 4% inulin fermented by *B. animalis*, **BbIM**—milk with 4% inulin and 0.06% sodium butyrate fermented by *B. animalis*, **CJ**—control milk fermented by *L. johnsonii*, **JI**—milk with 4% inulin fermented by *L. johnsonii*, **JIM**—milk with 4% inulin and 0.06% sodium butyrate fermented by *L. johnsonii*, **CA**—control milk fermented by *L. acidophilus*, **AI**—milk with 4% inulin fermented by *L. johnsonii*, **AIM**—milk with 4% inulin and 0.06% sodium butyrate fermented by *L. acidophilus*.

**Table 1 nutrients-17-02249-t001:** Milk groups obtained in experiment.

Bacterial Strain	Control Group	Group with Inulin 4%	Group with Inulin 4% and Sodium Butyrate 0.06%
*Lacticaseibacillus paracasei* L26	CP	PI	PIM
*Lacticaseibacillus casei* 431	CC	CI	CIM
*Bifidobacterium animalis* ssp. *Lactis* BB-12	CBb	BbI	BbIM
*Lactobacillus johnsonii* LJ	CJ	JI	JIM
*Lactobacillus acidophilus* LA-5	CA	AI	AIM

**Table 2 nutrients-17-02249-t002:** Effect of inulin and sodium butyrate on color parameters (L*, a*, b*, C*, h°) of milk fermented by probiotic bacteria.

Experimental Batch	L*	a*	b*	C*	h°
CP	92.57 ^d^ ± 0.67	−1.43 ^c^ ± 0.84	7.97 ^de^ ± 0.3	8.12 ^cd^ ± 0.34	101.71 ^cd^ ± 0.45
PI	90.72 ^e^ ± 1.11	−1.55 ^c^ ± 0.09	8.46 ^d^ ± 0.38	8.66 ^c^ ± 0.51	100.40 ^d^ ± 0.71
PIM	91.19 ^e^ ± 1.24	−1.71 ^c^ ± 0.07	8.29 ^d^ ± 0.37	8.45 ^c^ ± 0.37	101.67 ^d^ ± 0.41
CC	92.49 ^d^ ± 0.89	−1.55 ^c^ ± 0.08	7.86 ^e^ ± 0.48	8.00 ^cd^ ± 0.45	101.20 ^cd^ ± 0.9
CI	91.97 ^de^ ± 0.67	−1.54 ^c^ ± 0.14	7.74 ^e^ ± 0.49	7.88 ^d^ ± 0.49	101.03 ^d^ ± 0.76
CIM	91.45 ^e^ ± 0.97	−1.46 ^c^ ± 0.08	8.10 ^de^ ± 0.34	8.23 ^c^ ± 0.34	100.21 ^e^ ± 0.72
CBb	97.14 ^a^ ± 0.24	−2.58 ^b^ ± 0.09	11.28 ^a^ ± 0.33	11.57 ^a^ ± 0.32	102.85 ^c^ ± 0.62
BbI	94.40 ^c^ ± 0.07	−3.10 ^a^ ± 0.01	10.86 ^b^ ± 0.04	11.30 ^a^ ± 0.04	105.91 ^a^ ± 0.05
BbIM	96.19 ^a^ ± 0.57	−2.77 ^b^ ± 0.14	10.54 ^c^ ± 0.06	10.90 ^b^ ± 0.06	104.74 ^b^ ± 0.74
CJ	97.90 ^a^ ± 0.34	−2.27 ^b^ ± 0.16	10.70 ^c^ ± 0.12	10.94 ^b^ ± 0.12	101.97 ^cd^ ± 0.9
JI	95.61 ^b^ ± 0.63	−2.95 ^a^ ± 0.13	10.25 ^bc^ ± 0.24	10.66 ^b^ ± 0.27	106.05 ^a^ ± 0.37
JIM	96.63 ^a^ ± 0.42	−2.84 ^ab^ ± 0.15	10.62 ^c^ ± 0.08	10.99 ^b^ ± 0.1	104.99 ^b^ ± 0.71
CA	98.45 ^a^ ± 1.07	−1.89 ^c^ ± 0.48	11.36 ^a^ ± 0.28	11.52 ^a^ ± 0.21	99.47 ^e^ ± 1.55
AI	97.16 ^a^ ± 1.48	−2.23 ^b^ ± 0.77	10.59 ^bc^ ± 0.21	10.87 ^b^ ± 0.13	102.63 ^cd^ ± 1.22
AIM	95.87 ^ab^ ± 0.64	−2.72 ^b^ ± 0.41	10.58 ^c^ ± 0.08	10.95 ^b^ ± 0.06	105.35 ^ab^ ± 0.62

Mean ± standard deviation; ^a–e^—mean values denoted in columns by different letters differ statistically significantly at *p* ≤ 0.05. **CP**—control milk fermented by *L. paracasei*; **PI**—milk with 4% inulin fermented by *L. paracasei*, **PIM**—milk with 4% inulin and 0.06% sodium butyrate fermented by *L. paracasei*, **CC**—control milk fermented by *L casei*, **CI**—milk with 4% inulin fermented by *L. casei*, **CIM**—milk with 4% inulin and 0.06% sodium butyrate fermented by *L. casei*, **CBb**—control milk fermented by *B. animalis*, **BbI**—milk with 4% inulin fermented by *B. animalis*, **BbIM**—milk with 4% inulin and 0.06% sodium butyrate fermented by *B. animalis*, **CJ**—control milk fermented by *L. johnsonii*, **JI**—milk with 4% inulin fermented by *L. johnsonii*, **JIM**—milk with 4% inulin and 0.06% sodium butyrate fermented by *L. johnsonii*, **CA**—control milk fermented by *L. acidophilus*, **AI**—milk with 4% inulin fermented by *L. johnsonii*, **AIM**—milk with 4% inulin and 0.06% sodium butyrate fermented by *L. acidophilus.*

**Table 3 nutrients-17-02249-t003:** pH value, acidity, and syneresis of milk fermented by probiotic bacteria.

Experimental Batch	pH	TA%	Syneresis, %
CP	4.26 ^gh^ ± 0.02	0.93 ^ab^ ± 0.02	32.69 ^c^ ± 1.23
PI	4.23 ^h^ ± 0.01	0.88 ^c^ ± 0.02	26.34 ^d^ ± 1.38
PIM	4.25 ^g^ ± 0.01	0.90 ^b^ ± 0.02	23.96 ^e^ ± 1.06
CC	4.29 ^f^ ± 0.01	0.92 ^ab^ ± 0.03	28.53 ^d^ ± 1.82
CI	4.24 ^gh^ ± 0.01	0.89 ^c^ ± 0.01	25.72 ^e^ ± 1.22
CIM	4.28 ^f^ ± 0.01	0.93 ^b^ ± 0.01	20.43 ^f^ ± 1.77
CBb	4.63 ^a^ ± 0.01	0.76 ^f^ ± 0.01	41.30 ^a^ ± 1.54
BbI	4.51 ^b^ ± 0.03	0.79 ^e^ ± 0.01	40.7 ^a^ ± 1.73
BbIM	4.50 ^b^ ± 0.03	0.78 ^e^ ± 0.01	41.23 ^a^ ± 1.91
CJ	4.35 ^d^ ± 0.03	0.89 ^c^ ± 0.02	43.50 ^a^ ± 2.18
JI	4.33 ^e^ ± 0.01	0.96 ^a^ ± 0.01	26.07 ^d^ ± 1.41
JIM	4.31 ^de^ ± 0.03	0.92 ^b^ ± 0.02	27.97 ^d^ ± 1.3
CA	4.43 ^c^ ± 0.04	0.91 ^b^ ± 0.06	38.01 ^b^ ± 0.67
AI	4.37 ^d^ ± 0.03	0.94 ^ab^ ± 0.02	27.37 ^d^ ± 1.63
AIM	4.35 ^d^ ± 0.01	0.83 ^d^ ± 0.01	29.87 ^d^ ± 0.85

Mean ± standard deviation; ^a–h^—mean values denoted in columns by different letters differ statistically significantly at *p* ≤ 0.05. **CP**—control milk fermented by *L. paracasei*; **PI**—milk with 4% inulin fermented by *L. paracasei*, **PIM**—milk with 4% inulin and 0.06% sodium butyrate fermented by *L. paracasei*, **CC**—control milk fermented by *L. casei*, **CI**—milk with 4% inulin fermented by *L. casei*, **CIM**—milk with 4% inulin and 0.06% sodium butyrate fermented by *L. casei*, **CBb**—control milk fermented by *B. animalis*, **BbI**—milk with 4% inulin fermented by *B. animalis*, **BbIM**—milk with 4% inulin and 0.06% sodium butyrate fermented by *B. animalis*, **CJ**—control milk fermented by *L. johnsonii*, **JI**—milk with 4% inulin fermented by *L. johnsonii*, **JIM**—milk with 4% inulin and 0.06% sodium butyrate fermented by *L. johnsonii*, **CA**—control milk fermented by *L. acidophilus*, **AI**—milk with 4% inulin fermented by *L. johnsonii*, **AIM**—milk with 4% inulin and 0.06% sodium butyrate fermented by *L. acidophilus.*

**Table 4 nutrients-17-02249-t004:** Effect of inulin and sodium butyrate on organoleptic parameters of milk fermented by probiotic bacteria.

Experimental Batch	Consistency	Milky-Creamy Taste	Sour Taste	Sweet Taste	Off-Taste	Sour Odor	Off-Odor
CP	7.33 ^bc^ ± 0.78	6.67 ^a^ ± 0.66	4.25 ^c^ ± 0.76	5.58 ^a^ ± 0.64	1.42 ^a^ ± 0.40	3.58 ^b^ ± 0.07	1.00 ^a^ ± 0.00
PI	6.92 ^c^ ±0.78	5.33 ^b^ ± 0.27	5.50 ^b^ ± 0.83	3.92 ^bc^ ± 0.47	1.50 ^a^ ± 0.50	3.42 ^b^ ± 0.35	1.00 ^a^ ± 0.00
PIM	7.08 ^c^ ± 0.44	5.22 ^b^ ± 0.83	5.83 ^b^ ± 0.76	4.67 ^b^ ± 0.42	1.25 ^a^ ± 0.25	5.58 ^a^ ± 0.56	1.50 ^a^ ± 0.50
CC	7.87 ^b^ ± 0.25	6.53 ^a^ ± 0.50	4.67 ^c^ ± 0.53	4.40 ^c^ ± 0.35	1.67 ^a^ ± 0.64	3.27 ^b^ ± 0.87	1.00 ^a^ ± 0.00
CI	7.73 ^b^ ± 0.33	5.27 ^b^ ± 0.35	5.87 ^b^ ± 0.29	3.73 ^c^ ± 0.50	1.13 ^a^ ± 0.13	3.33 ^b^ ± 0.59	1.00 ^a^ ± 0.00
CIM	7.13 ^bc^ ± 0.92	5.27 ^b^ ± 0.34	6.00 ^a^ ± 0.93	3.47 ^d^ ± 0.55	1.67 ^a^ ± 0.60	4.67 ^a^ ± 1.09	1.47 ^a^ ± 0.70
CBb	6.80 ^c^ ± 0.84	5.20 ^b^ ± 0.43	4.80 ^c^ ± 0.84	5.20 ^a^ ± 0.89	1.00 ^a^ ± 0.00	2.60 ^c^ ± 0.14	1.00 ^a^ ± 0.00
BbI	6.40 ^d^ ± 0.55	5.40 ^b^ ± 0.67	5.20 ^b^ ± 0.52	4.80 ^b^ ± 0.59	1.00 ^a^ ± 0.00	2.80 ^c^ ± 0.55	1.00 ^a^ ± 0.00
BbIM	6.20 ^d^ ± 0.30	5.00 ^bc^ ± 0.58	5.40 ^b^ ± 0.61	4.40 ^c^ ± 0.90	1.40 ^a^ ± 0.40	3.60 ^b^ ± 0.52	1.00 ^a^ ± 0.00
CJ	7.75 ^b^ ± 0.500	4.50 ^c^ ± 0.91	4.75 ^c^ ± 0.31	6.25 ^a^ ± 0.92	1.25 ^a^ ± 0.25	2.25 ^c^ ± 0.50	1.00 ^a^ ± 0.00
JI	8.25 ^ab^ ± 0.96	4.50 ^c^ ± 2.08	4.25 ^c^ ± 0.80	5.25 ^a^ ± 0.81	1.25 ^a^ ± 0.25	2.00 ^c^ ± 0.15	1.00 ^a^ ± 0.00
JIM	8.75 ^a^ ± 0.50	5.00 ^bc^ ± 0.58	5.25 ^b^ ± 0.50	5.00 ^ab^ ± 0.71	1.75 ^a^ ± 0.75	3.50 ^b^ ± 0.58	1.00 ^a^ ± 0.00
CA	7.13 ^c^ ± 0.23	4.93 ^bc^ ± 0.83	4.60 ^c^ ± 0.45	4.40 ^b^ ± 0.23	1.67 ^a^ ± 0.60	3.27 ^b^ ± 0.87	1.00 ^a^ ± 0.00
AI	6.80 ^c^ ± 0.50	4.67 ^c^ ± 0.02	4.80 ^c^ ± 0.90	4.00 ^c^ ± 0.24	1.60 ^a^ ± 0.50	3.53 ^b^ ± 0.92	1.00 ^a^ ± 0.00
AIM	6.13 ^d^ ± 0.45	4.33 ^c^ ± 0.59	6.00 ^a^ ± 0.67	3.80 ^c^ ± 0.21	1.80 ^a^ ± 0.80	4.93 ^a^ ± 0.31	1.50 ^a^ ± 0.50

Mean ± standard deviation; ^a–d^—mean values denoted in columns by different letters differ statistically significantly at *p* ≤ 0.05. **CP**—control milk fermented by *L. paracasei*; **PI**—milk with 4% inulin fermented by *L. paracasei*, **PIM**—milk with 4% inulin and 0.06% sodium butyrate fermented by *L. paracasei*, **CC**—control milk fermented by *L. casei*, **CI**—milk with 4% inulin fermented by *L. casei*, **CIM**—milk with 4% inulin and 0.06% sodium butyrate fermented by *L. casei*, **CBb**—control milk fermented by *B. animalis*, **BbI**—milk with 4% inulin fermented by *B. animalis*, **BbIM**—milk with 4% inulin and 0.06% sodium butyrate fermented by *B. animalis*, **CJ**—control milk fermented by *L. johnsonii*, **JI**—milk with 4% inulin fermented by *L. johnsonii*, **JIM**—milk with 4% inulin and 0.06% sodium butyrate fermented by *L. johnsonii*, **CA**—control milk fermented by *L. acidophilus*, **AI**—milk with 4% inulin fermented by *L. johnsonii*, **AIM**—milk with 4% inulin and 0.06% sodium butyrate fermented by *L. acidophilus.*

**Table 5 nutrients-17-02249-t005:** Viable counts of bacteria (log cfu g^−1^) in milk fermented by probiotic bacteria before digestion, in the oral cavity, stomach, and small intestine.

Experimental Batches	Viable Counts of Probiotic Bacteria, log CFU g^−1^			
	Digestion Phase			
	Before Digestion	Oral Cavity	Stomach	Small Intestine
CP	10.50 ^eA^ ± 0.29	10.48 ^dA^ ± 0.14	3.73 ^fC^ ± 0.12	5.01 ^fB^ ± 0.05
PI	10.45 ^eA^ ± 0.23	10.43 ^dA^ ± 0.22	4.03 ^eC^ ± 0.25	6.40 ^cB^ ± 0.03
PIM	10.50 ^eA^ ± 0.14	10.52 ^dA^ ± 0.21	3.46 ^fC^ ± 0.32	5.01 ^fB^ ± 0.19
CC	11.06 ^cA^ ± 0.04	11.00 ^bA^ ± 0.14	1.43 ^hC^ ± 0.51	5.17 ^fB^ ± 0.15
CI	11.22 ^bA^ ± 0.06	11.16 ^bA^ ± 0.10	2.87 ^gC^ ± 0.08	5.45 ^eB^ ± 0.40
CIM	10.70 ^dA^ ± 0.08	10.70 ^cdA^ ± 0.16	2.80 ^gC^ ± 0.04	5.88 ^eB^ ± 0.70
CBb	10.78 ^dA^ ± 0.14	10.53 ^dA^ ± 0.12	6.23 ^dC^ ± 0.04	6.50 ^cB^ ± 0.16
BbI	10.63 ^dA^ ± 0.20	10.38 ^eA^ ± 0.28	6.55 ^cC^ ± 0.05	7.15 ^aB^ ± 0.22
BbIM	10.84 ^dA^ ± 0.41	10.68 ^cdA^ ± 0.08	6.40 ^cB^ ± 0.11	6.54 ^cB^ ± 0.25
CJ	11.97 ^aA^ ± 0.16	11.84 ^aA^ ± 0.08	6.49 ^cC^ ± 0.07	7.51 ^aB^ ± 0.29
JI	11.95 ^aA^ ± 0.07	11.04 ^bA^ ± 0.19	6.50 ^cC^ ± 0.01	7.32 ^aB^ ± 0.12
JIM	10.94 ^cA^ ± 0.16	10.76 ^cA^ ± 0.13	6.74 ^bB^ ± 0.03	6.89 ^bB^ ± 0.11
CA	10.88 ^dA^ ± 0.09	10.81 ^cA^ ± 0.18	4.44 ^eC^ ± 0.05	6.06 ^dB^ ± 0.07
AI	10.80 ^dA^ ± 0.10	10.33 ^eA^ ± 0.35	4.73 ^eC^ ± 0.06	7.42 ^aB^ ± 0.14
AIM	10.98 ^cdA^ ± 0.36	10.94 ^cA^ ± 0.08	6.85 ^aB^ ± 0.05	6.87 ^bB^ ± 0.09

Mean ± standard deviation; ^a–h^—mean values denoted in columns by different letters differ statistically significantly at *p* ≤ 0.05; ^A–C^—mean values in rows denoted by different letters differ significantly at *p* ≤ 0.05. **CP**—control milk fermented by *L. paracasei*; **PI**—milk with 4% inulin fermented by *L. paracasei*, **PIM**—milk with 4% inulin and 0.06% sodium butyrate fermented by *L. paracasei*, **CC**—control milk fermented by *L. casei*, **CI**—milk with 4% inulin fermented by *L. casei*, **CIM**—milk with 4% inulin and 0.06% sodium butyrate fermented by *L. casei*, **CBb**—control milk fermented by *B. animalis*, **BbI**—milk with 4% inulin fermented by *B. animalis*, **BbIM**—milk with 4% inulin and 0.06% sodium butyrate fermented by *B. animalis*, **CJ**—control milk fermented by *L. johnsonii*, **JI**—milk with 4% inulin fermented by *L. johnsonii*, **JIM**—milk with 4% inulin and 0.06% sodium butyrate fermented by *L. johnsonii*, **CA**—control milk fermented by *L. acidophilus*, **AI**—milk with 4% inulin fermented by *L. johnsonii*, **AIM**—milk with 4% inulin and 0.06% sodium butyrate fermented by *L. acidophilus.*

## Data Availability

The original contributions presented in this study are included in the article. The raw data supporting the conclusions—including unprocessed measurement results—are available from the corresponding authors upon reasonable request (aznamirowska@ur.edu.pl, kszajnar@ur.edu.pl).

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
