# Peer review of "Fermented Milk Supplemented with Sodium Butyrate and Inulin: Physicochemical Characterization and Probiotic Viability Under In Vitro Simulated Gastrointestinal Digestion"

_nutrients, 2025, doi:10.3390/nu17132249_

Round 1
Reviewer 1 Report
Comments and Suggestions for Authors
The objectives of the study should be reformulated to include the investigation described in the abstract to include physicochemical and sensory evaluations? The protocol for in vitro digestion is not very clear. For example, “Next, the gastric phase was initiated by adding pepsin at a concentration of 13.08 142 mg per sample.” This means that the enzyme was directly added to the 50 mL of sample after oral digestion? More details are needed.
The authors suggested changes in color parameters according to the strain used. How the authors explain this phenomenon. In Table 5, please correct to Small Intestine.
How the authors explain the significant decrease of cfu in gastric environment, followed by an increase in small intestine?
The authors are asking to provide the statements for ethical approval regarding sensorial analysis.
Author Response
Reviewer 1
The Authors thank you for performing the review. We have responded to the comments:
The objectives of the study should be reformulated to include the investigation described in the abstract to include physicochemical and sensory evaluations?
L15:
This study aimed to evaluate the effects of milk supplementation with inulin and sodium butyrate on physicochemical properties, sensory characteristics, and the survival of selected probiotic strains during in vitro simulated gastrointestinal digestion.
L145:
In light of accumulating evidence on sodium butyrate–probiotic–prebiotic interactions, the present study aims to evaluate the effects of sodium butyrate and inulin supplementation in milk on the physicochemical properties and sensory attributes of probiotic-fermented milk, and assess the survival of selected probiotic strains under simulated gastrointestinal conditions.
The protocol for in vitro digestion is not very clear. For example, “Next, the gastric phase was initiated by adding pepsin at a concentration of 13.08 142 mg per sample.” This means that the enzyme was directly added to the 50 mL of sample after oral digestion? More details are needed.
L231:
A static in vitro gastrointestinal digestion model (oral, gastric, intestinal phases) was employed based on established protocols [12,13], with minor modifications.
For each formulation of fermented milk, three digestion experiments were conducted in parallel, each using an independent 50 mL aliquot. Each aliquot was processed up to a defined phase—oral, gastric, or intestinal—and terminated accordingly for subsequent analysis. All digestion phases were performed in triplicate per formulation to ensure reproducibility.
During the oral phase, each 50 mL aliquot of fermented milk was combined with 5 mL of simulated salivary fluid in a 100 mL amber glass vessel. The simulated salivary fluid was prepared by dissolving 2.38 g Na₂HPO₄, 0.19 g K₂HPO₄, 8.0 g NaCl, 100 mg mucin, and 150 mg α-amylase (200 U/L activity) in 1 L of deionized water. The mixtures were adjusted to pH 6.75 ± 0.20 with 1 M NaOH and incubated at 37 °C for 10 min with constant stirring (90 rpm) to simulate oral digestion.
Gastric digestion was then initiated by adding pepsin (13.08 mg per 50 mL aliquot) to each oral bolus and acidifying the mixture to pH 2.00 ± 0.20 with 12 M HCl. The samples were incubated at 37 °C for 2 h under constant stirring (90 rpm) to simulate gastric conditions.
Finally, to simulate the intestinal phase, 5 mL of pancreatin solution (4 g/L in deionized water) and 5 mL of bile salt solution (25 g/L) were added to each gastric digest. The pH was raised to 7.00 ± 0.20 (using 1 M NaOH), and incubation was continued at 37 °C for an additional 2 h with constant stirring (90 rpm).
Each digestion was halted at its designated end point (oral, gastric, or intestinal phase), and the resulting digesta were collected immediately for microbiological analysis as described in Section 2.8.
The authors suggested changes in color parameters according to the strain used. How the authors explain this phenomenon.
L305:
The strain-dependent differences in color parameters observed in our study can be attributed to variations in fermentation metabolism, particularly in acid production and its effects on the milk matrix. Statistical analysis (Table 2) confirmed that both strain type and supplementation significantly influenced (p < 0.05) the color parameters L* and b*. Specifically, fermented milks inoculated with B. animalis, L. johnsonii, and L. acidophilus—which produced higher terminal pH values (4.5)—exhibited significantly greater lightness (L*) and yellow hue (b*) compared to those fermented by more strongly acidifying strains such as L. casei and L. paracasei, which reached pH values of 4.2–4.3. These latter samples appeared darker (lower L*) and less yellow (lower b*).
This pattern aligns with established relationships between pH, casein aggregation, and visual appearance. As the pH of fermenting milk approaches the isoelectric point of casein (4.6), protein precipitation and whey separation (syneresis) intensify, reducing the brightness and increasing the translucency of the fermented matrix [40,41]. Conversely, a slightly higher pH results in a looser curd structure and greater whey retention, leading to a lighter, more opaque appearance.
In addition, we hypothesize that differences in exopolysaccharide (EPS) production, although not directly measured in this study, may have contributed to the observed color variation. Strains such as L. acidophilus and B. animalis are known to produce EPS, which enhance water-holding capacity and increase light scattering by creating a porous protein matrix [41]. Conversely, strains with lower EPS production—such as L. casei and L. paracasei—likely formed denser curds with more whey separation, resulting in a darker and less turbid appearance. According to Cais Sokolińska and Pikul [42], the microbiological activity of yogurt microflora leads to the changes in potential acidity of the clot, and thus cause physical changes in the clot, resulting in a difference in the color brightness and the degree of color saturation. These physiological and metabolic traits, together with acidification profiles, explain the statistically significant differences in color parameters observed among probiotic fermented milk samples.
In Table 5, please correct to Small Intestine.
We have corrected the term “small intensine” to “small intestine” in the Table 5 caption of the revised manuscript.
How the authors explain the significant decrease of cfu in gastric environment, followed by an increase in small intestine?
L470:
The pronounced reduction in viable counts during the simulated gastric phase can be attributed to the synergistic lethality of an extremely low pH (≈2) and the proteolytic activity of pepsin, which compromise bacterial envelope integrity and denature essential enzymes, causing many cells to become non-culturable or to die outright [53]. A significant fraction of the population, however, survives as sub-lethally injured cells: although unable to form colonies under acute acid stress, these cells retain the capacity for repair and resuscitation once they encounter milder conditions. Upon transition to the small-intestinal simulation (where pH is neutralized to 7.0 and pancreatin plus bile salts exert milder, more selective pressure) these injured cells can restore membrane integrity, reactivate stress-response systems (e.g., chaperone proteins and proton-pumping F₀F₁-ATPases) and resume cell division, leading to an apparent “rebound” in CFU [53,54]. Moreover, many Lactobacillus and Bifidobacterium strains express bile salt hydrolases (BSHs) that deconjugate and detoxify bile acids, further enhancing survival and modest growth in the intestinal milieu [55]. Finally, the presence of prebiotic substrates such as inulin provides fermentable carbon sources and may exert an osmoprotective effect, facilitating bacterial recovery and limited proliferation during the two-hour intestinal incubation [56].
The authors are asking to provide the statements for ethical approval regarding sensorial analysis.
Please find attached the templates of the informed consent and ethical approval statements in Polish (all members of the organoleptic evaluation panel were Polish) along with their English translations.
The conducted organoleptic tests involved standard methods of sensory evaluation, in which participants consumed safe food products made from ingredients approved for human consumption. Additionally, the organoleptic evaluation did not pose any health risks to the participants. Therefore, such studies did not require approval from the Ethics Committee or Institutional Review Board.
We supplemented the conclusion with the following passage: L551
Both inulin and the inulin–butyrate combination induced a perceptible darkening of the fermented milk samples compared to their respective controls, regardless of the probiotic strain employed. Notably, the addition of either inulin or the inulin–butyrate combination led to a significant intensification of green hue only in the L. acidophilus groups. Inclusion of the inulin-sodium butyrate combination led to a significant pH reduction only in the milk fermented by B. animalis and L. acidophilus. Inulin and the combined inulin–butyrate supplementation significantly reduced syneresis in milk fermented by L. johnsonii, L. acidophilus, L. casei, and L. paracasei compared to their control counterparts. In an organoleptic evaluation, no unpleasant taste or odor was detected in any sample, even despite the addition of sodium butyrate.

Reviewer 2 Report
Comments and Suggestions for Authors
I have carefully reviewed the manuscript entitled “Sodium Butyrate–Enriched Probiotic Fermented Milk: Characterization of the Fermented Matrix and Viability of Probiotic Bacteria under Simulated Gastrointestinal Conditions.”
Below, I outline the most significant observations and concerns regarding the study.
Title and Abstract:
The title and abstract require improvement for clarity and accuracy.
- While the manuscript (including the Abstract, Materials and Methods, and Results sections) refers to the addition of inulin, the title does not mention this component. I strongly recommend including inulin in the title to reflect the full scope of the study.
- I suggest replacing the phrase “under simulated gastrointestinal conditions” with the more precise terminology: “under in vitro simulated gastrointestinal digestion.” This better reflects the nature of the experimental setup.
- The use of the term “matrix” in the current title is problematic. In food science, matrix typically refers to the structural and compositional organization of a food system, encompassing interactions between proteins, fats, carbohydrates, and water. However, in this study, the parameters assessed (e.g., acidity, syneresis, and color) do not support such a comprehensive characterization. Therefore, I recommend either removing the term or replacing it with a more appropriate descriptor.
- I also believe the use of the term “Probiotic Fermented Milk” in the title is inappropriate. The term probiotic should be reserved for products or strains that meet established probiotic criteria (e.g., health benefit supported by scientific evidence). Unless the final product has been proven to exert a probiotic effect, or the strains used are officially recognized as probiotics at the applied dose, it is more accurate to refer to the “fermented milk with probiotic strains” or similar wording.
Introduction – Probiotic Strains:
- The authors should provide supporting data and references regarding the acidification capacity (i.e., lactic acid production) of each probiotic strain studied. This is particularly important since the strains were grown as single cultures, rather than in combination with a standard lactic acid-producing starter culture typically used in fermented dairy products. Clarifying this aspect will help assess the relevance and functionality of each strain in the fermentation process.
- The manuscript should also include references and data related to the gastric acid and bile salt tolerance of the probiotic strains used. These are critical parameters in establishing the viability and potential probiotic efficacy of the strains under gastrointestinal conditions.
- were the growth conditions (e.g., temperature at 37°C) uniform across all probiotic strains?. This should be explicitly stated, as variations in optimal growth conditions could affect the interpretation of results and comparability between strains.
Inulin and sodium butyrate supplements:
Use of Inulin and Sodium Butyrate – Literature References: The manuscript lacks sufficient and relevant literature references regarding the use of inulin and sodium butyrate as supplements in probiotic fermented milk.
- The authors should specifically address the role of sodium butyrate in promoting microbial growth, and provide appropriate references on its impact on the viability, growth, and metabolic activity of probiotic strains — particularly the same or similar strains investigated in this study.
- Relevant scientific evidence and citations should also be included to support the use of inulin and sodium butyrate as functional ingredients aimed at enhancing probiotic viability and contributing to gut health benefits. This background is essential to justify their inclusion in the formulation.
- Some of the references cited by the authors do not appear to be aligned with the statements they are intended to support. For instance, reference 4 and 5 (line 61) do not correspond to the content described in lines 59–61. The authors should carefully verify the relevance and accuracy of all cited sources.
Materials and Methods
Raw Material and Experimental Design – General Comments:
The overall experimental design of the study presents several weaknesses, which are also reflected in the quality and interpretation of the results.
- The milk used in the experiments was obtained from (Łaciate, SM Mlekpol, Grajewo, Poland). The precise composition of this milk should be clearly stated (e.g., fat content, protein, lactose, etc.), and confirmation of the absence of antibiotics is important, as antibiotic residues could inhibit microbial growth and compromise the validity of the results.
- A comprehensive compositional analysis of the milk is necessary, including at minimum: fat, protein, lactose, and ash content. This would allow for better interpretation of both the fermentation process and the behavior of the probiotic strains.
- The volume of milk used in each experiment and for each fermentation is not specified. This information is essential, as the quantities must be sufficient to perform the full set of physicochemical, microbiological, and sensory analyses.
- The manuscript does not report the initial concentration (CFU/mL) of each probiotic strain used at the start of fermentation. Additionally, no references or descriptions of the microbiological methods used for strain enumeration and identification are provided.
- The medium or substrate used to activate the probiotic strains at 40°C prior to inoculation into milk is not described. This information is important for assessing the reproducibility and viability of the strains before fermentation.
- Regarding titratable acidity (TA) (line 100), the manuscript states: “Titratable acidity (TA), expressed as grams of lactic acid per liter of milk…” However, the calculation method appears to be incorrect, as the reported values in Table 3 do not align with expected lactic acid concentrations for fermented milk. The authors should review and clarify both the method of calculation and the interpretation of the results.
Conclusion:
Given these shortcomings, I cannot approve the manuscript for publication. I therefore recommend that this article is unsuitable for publication in this journal.
Author Response
The Authors thank you for performing the review. We have responded to the comments:
Reviewer 2
Title and Abstract:
The title and abstract require improvement for clarity and accuracy.
We appreciate the reviewer’s detailed feedback on the clarity and scientific accuracy of the title and abstract. In response, we have revised the title to fully reflect the scope and nature of the study, as well as to comply with terminology best practices in food science.
L2: New title: Fermented Milk Supplemented with Sodium Butyrate and Inulin: Physicochemical Characterization and Probiotic Viability under In Vitro Simulated Gastrointestinal Digestion
Introduction – Probiotic Strains:
The authors should provide supporting data and references regarding the acidification capacity (i.e., lactic acid production) of each probiotic strain studied. This is particularly important since the strains were grown as single cultures, rather than in combination with a standard lactic acid-producing starter culture typically used in fermented dairy products. Clarifying this aspect will help assess the relevance and functionality of each strain in the fermentation process.
L101: All probiotic strains used in this study are capable of fermenting milk as monocultures by converting lactose into lactic acid and lowering the pH. Lactobacillus acidophilus is well established as a traditional “acidophilus milk” culture, owing to its homofermentative metabolism that drives lactic acid production and significant pH reduction in milk-based systems [14]. Although L. acidophilus grows more slowly than standard yogurt starters, studies confirm its ability to acidify milk to pH values between 4.0 and 4.6 over extended fermentation periods [15]. Lactobacillus casei also possesses strong acidifying potential and has been widely used as a sole fermentative agent in dairy products such as fermented milks and probiotic drinks [16]. Its facultative heterofermentative metabolism allows for efficient lactic acid production under appropriate conditions, typically resulting in final pH values below 4.5 [17]. Bifidobacterium animalis subsp. lactis, while generally less acidifying than the aforementioned lactobacilli, also demonstrates the ability to ferment milk substrates. Pure cultures of B. animalis HN019 were able to reduce milk pH from 6.0 to approximately 5.3 within 14 hours of fermentation [18]. Though slower to acidify, B. animalis contributes to pH reduction and metabolite production, especially when grown under anaerobic conditions or in combination with prebiotics.
The manuscript should also include references and data related to the gastric acid and bile salt tolerance of the probiotic strains used. These are critical parameters in establishing the viability and potential probiotic efficacy of the strains under gastrointestinal conditions.
L118: Probiotic strains must overcome the hostile conditions of the human gastrointestinal tract - namely, gastric acidity (pH < 3.0) and bile salts (0.2 - 2.0 % w/v) - in order to survive transit and exert health‐promoting effects. Vernazza et al. [19] demonstrated that Bifidobacterium animalis ssp. lactis BB-12 suspended in glycine–HCl buffer (pH 2.0; 37 °C) retained > 99 % viability during the first 20 s, with only modest declines over longer exposures, whereas at pH 4.0 it maintained 81 % viability after 20 min. In bile - salt assays (0.5 % Oxgall; 37 °C; 2 h), BB-12 reached 1.22 × 108 CFU/mL versus 7.65 × 108 CFU/mL in bile - free controls - equating to 16 % survival - despite negligible changes in optical density, a phenomenon attributed to bile-induced cell autoaggregation [19]. Jacobsen et al. [20] showed that Lactobacillus casei 431® retained > 10⁵ CFU/mL when exposed at pH 2.5 for 2 h followed by 0.3 % Oxgall for 4 h (a < 2 log₁₀ reduction from 10⁸ CFU/mL). Salgaço et al. [21] found that L. acidophilus LA-5 suffered a 3.6 log₁₀ CFU/mL reduction after 2 h in simulated gastric juice (0.3 % HCl; pepsin 1 g/L; pH 2.0; 37 °C) and an additional 3.8 log₁₀ loss after 4 h in simulated intestinal fluid (0.3 % bile salts; pancreatin 1 g/L; pH 7.4; 37 °C), whereas micro-encapsulation in an ultra-filtered cheese matrix preserved viability at 6 log₁₀ CFU/g through the full 6 h sequence [22] and planktonic cells tolerated 0.3 % bile for 4 h with < 1 log₁₀ CFU/mL reduction [23]. In vitro assessment of L. paracasei L26 indicated a decline from 8.96 to 5.11 log₁₀ CFU/mL (3.85 log drop) after 3 h at pH 2.0 (0.3 % HCl; pepsin 1 g/L; 37 °C) and survival of 6.30 log₁₀ CFU/mL at pH 3.0 (2.66 log drop), while exposure to 0.5 % and 1.0 % Oxgall for 6 h resulted in < 1 log₁₀ CFU loss from > 108 CFU/mL [24]. Despite its commercial use, no peer - reviewed data exist on the acid or bile tolerance of Lactobacillus johnsonii LJ. Available studies refer to other isolates (e.g. N5, N7, PF01), which survive pH 2.5 and 0.3 % bile at rates of 33 %–114 % over 2–4 h [25].
Were the growth conditions (e.g., temperature at 37°C) uniform across all probiotic strains?. This should be explicitly stated, as variations in optimal growth conditions could affect the interpretation of results and comparability between strains.
Yes, the growth conditions (e.g., temperature at 37°C) were uniform across all probiotic strains.
Inulin and sodium butyrate supplements:
Use of Inulin and Sodium Butyrate – Literature References: The manuscript lacks sufficient and relevant literature references regarding the use of inulin and sodium butyrate as supplements in probiotic fermented milk.
- The authors should specifically address the role of sodium butyrate in promoting microbial growth, and provide appropriate references on its impact on the viability, growth, and metabolic activity of probiotic strains — particularly the same or similar strains investigated in this study.
L91: Sodium butyrate, as a principal SCFA, represents a key microbial metabolite with a broad spectrum of intestinal health–promoting activities. It fosters microbial diversity and supports the growth of beneficial bacteria while suppressing pathogenic species [13].Although sodium butyrate is not directly utilized as a growth substrate by Lactobacillus casei or Bifidobacterium, it creates a favorable colonic environment by attenuating inflammation, reinforcing the intestinal barrier and modulating luminal pH. Direct fortification of foods—including milk—with sodium butyrate is challenged by its characteristic odor; however, co-administration of inulin and sodium butyrate alongside probiotics in a fermented milk matrix yields a synbiotic formulation that may confer synergistic health benefits.
L71: Beyond its nutritive role, butyrate acts as a signaling mediator, modulating diverse cellular functions in the gut and peripheral tissues. Sodium butyrate markedly shapes the gut microenvironment by fortifying epithelial integrity, exerting anti-inflammatory effects—reducing pro-inflammatory cytokine production (e.g., IL-1β, IL-6, TNF-α), inhibiting NF-κB activation and modulating immune cell function—and by lowering luminal pH. Collectively, these mechanisms foster an ecosystem conducive to the growth and activity of beneficial bacteria, thereby supporting overall intestinal and immunological homeostasis [7].
- Relevant scientific evidence and citations should also be included to support the use of inulin and sodium butyrate as functional ingredients aimed at enhancing probiotic viability and contributing to gut health benefits. This background is essential to justify their inclusion in the formulation.
L79:There is a growing body of evidence demonstrating that inulin (a prebiotic) and sodium butyrate (a short-chain fatty acid) exert beneficial effects on gut health and can enhance the viability and functionality of probiotic strains. Inulin is a water-soluble oligosaccharide (a fructan) that can be extracted from various sources, including chicory, garlic, wheat, oats and bulgur [8]. Structurally, it consists of β-(2→1)-linked fructosyl units terminating in a glucosyl residue [9]. As a recognized prebiotic, inulin is increasingly incorporated into fermented products such as yogurt to improve gastrointestinal health, enhance calcium absorption and support immune function. Moreover, it stimulates the proliferation of probiotic genera—such as Lactobacillus and Bifidobacterium—during storage, thereby ensuring delivery of high numbers of viable cells to the colon [10]. Inulin has also been shown to improve the physicochemical, functional and sensory properties of dairy matrices, while preserving probiotic viability in products like yogurt [11,12].
Sodium butyrate, as a principal SCFA, represents a key microbial metabolite with a broad spectrum of intestinal health–promoting activities. It fosters microbial diversity and supports the growth of beneficial bacteria while suppressing pathogenic species [13].Although sodium butyrate is not directly utilized as a growth substrate by Lactobacillus casei or Bifidobacterium, it creates a favorable colonic environment by attenuating inflammation, reinforcing the intestinal barrier and modulating luminal pH. Direct fortification of foods—including milk—with sodium butyrate is challenged by its characteristic odor; however, co-administration of inulin and sodium butyrate alongside probiotics in a fermented milk matrix yields a synbiotic formulation that may confer synergistic health benefits.3. Some of the references cited by the authors do not appear to be aligned with the statements they are intended to support. For instance, reference 4 and 5 (line 61) do not correspond to the content described in lines 59–61. The authors should carefully verify the relevance and accuracy of all cited sources.
This text and its citations have been removed.
Materials and Methods
Raw Material and Experimental Design – General Comments:
The milk used in the experiments was obtained from (Łaciate, SM Mlekpol, Grajewo, Poland). The precise composition of this milk should be clearly stated (e.g., fat content, protein, lactose, etc.), and confirmation of the absence of antibiotics is important, as antibiotic residues could inhibit microbial growth and compromise the validity of the results.
A comprehensive compositional analysis of the milk is necessary, including at minimum: fat, protein, lactose, and ash content. This would allow for better interpretation of both the fermentation process and the behavior of the probiotic strains.
Thank you for your suggestion to supplement the milk composition data. In the revised manuscript (Materials and Methods) we have included only the values declared on the product label:
L154:Milk used in the experiments (Łaciate, SM Mlekpol, Grajewo, Poland) had the following composition per 100 mL: fat 2.0 g, lactose 4.8 g, protein: 3.3 g, salt 0.10 g, calcium: 120 mg.
At the same time, we wish to clarify that this commercially available pasteurized milk is authorized for sale in the European Union and is subject to mandatory quality controls—including maximum residue limits for veterinary drugs (including antibiotics) under Commission Regulation (EU) No 37/2010 and detailed hygiene rules for food of animal origin under Regulation (EC) No 853/2004. Accordingly, the producer guarantees that any antibiotic residues fall within permissible limits and cannot affect the growth of probiotic cultures.
The volume of milk used in each experiment and for each fermentation is not specified. This information is essential, as the quantities must be sufficient to perform the full set of physicochemical, microbiological, and sensory analyses.
Thank you for this important observation. For each experimental condition, we prepared a 2 000 mL batch of milk (control, inulin-fortified, and inulin + sodium butyrate–fortified) which was then inoculated and aliquoted into twenty 100 mL polypropylene containers (n = 20; total 2 000 mL per condition). This volume ensures ample material for all downstream assays. 50 mL of sample was used for simulated gastrointestinal digestion, and the remaining volume was allocated to physicochemical measurements, microbiological analysis, and organoleptic evaluation.
L177: For each experimental condition, a 2 000 mL batch of inoculated milk (control, inulin-fortified, and inulin + sodium butyrate–fortified) was dispensed into twenty 100 mL polypropylene containers (n = 20) to provide sufficient material for all analyses and in vitro digestion.
The manuscript does not report the initial concentration (CFU/mL) of each probiotic strain used at the start of fermentation. Additionally, no references or descriptions of the microbiological methods used for strain enumeration and identification are provided.
We apologize for the omission. After the 5 h activation step at 40 °C, each probiotic culture reached approximately log 9 CFU g⁻¹. This inoculum was then added to the milk at 5 % (w/w). Enumeration and identification of the probiotic strains were performed by the same microbiological methods described in Section 2.4.
L171: After cooling to 37 ± 1 °C, the milk (Łaciate, SM Mlekpol, Grajewo, Poland) was inoculated (5 %, w/w) with a single, active probiotic strain (activated to ~log 9 CFU g⁻¹)—either B. animalis BB-12, L. johnsonii LJ, L. casei 431, L. paracasei L26, or L. acidophilus LA-5—following a 5 h activation step at 40 °C.
The medium or substrate used to activate the probiotic strains at 40°C prior to inoculation into milk is not described. This information is important for assessing the reproducibility and viability of the strains before fermentation.
Prior to inoculation, all probiotic strains were activated in the same pasteurized milk (Łaciate, SM Mlekpol, Grajewo, Poland). Activation was carried out at 40 °C for 5 h in this milk to ensure strain viability and reproducibility before fermentation.
L169: Each milk batch (control, inulin-fortified, and inulin + sodium butyrate–fortified) was homogenized at 60 °C under 20 MPa, then re-pasteurized at 85 °C for 10 minutes. After cooling to 37 ± 1 °C, the milk (Łaciate, SM Mlekpol, Grajewo, Poland) was inoculated (5 %, w/w) with a single, active probiotic strain (activated to ~log 9 CFU g⁻¹)—either B. animalis BB-12, L. johnsonii LJ, L. casei 431, L. paracasei L26, or L. acidophilus LA-5—following a 5 h activation step at 40 °C.
Regarding titratable acidity (TA) (line 100), the manuscript states: “Titratable acidity (TA), expressed as grams of lactic acid per liter of milk…” However, the calculation method appears to be incorrect, as the reported values in Table 3 do not align with expected lactic acid concentrations for fermented milk. The authors should review and clarify both the method of calculation and the interpretation of the results.
L:189 The acidity of milk was analysed by titratable method using 0,1N NaOH solution and phenolphthalein as indicator [27]. The volume of NaOH required to neutralize milk acids was recorded and used to calculate the content of titratable acids eferring to the following equation:
Total acidity (g of lactic acid equivalent/l) = (VNaOH*0,009) / (VSample*0,01)
Where VNaOH is the volume of NaOH used to neutralize the lactic acid (ml); 0.009 conversion
factor whereby 1 ml NaOH (0.01 N) neutralizes 0.009 g of lactic acid; V Sample is the volume
of milk sample for titration (ml); and 0,01 conversion factor from % of acidity to g of lactic
acid equivalent/l.
Our total acidity results are comparable to those reported by other authors, for example Kim et al. (2020), who obtained similar total acidity values in yogurt.
Conclusion:
Given these shortcomings, I cannot approve the manuscript for publication. I therefore recommend that this article is unsuitable for publication in this journal.
The authors would like to thank the Reviewer for the insightful review, and we hope that our revisions will lead to a reversal of the decision and to acceptance of the manuscript for publication in Nutrients.
Round 2
Reviewer 1 Report
Comments and Suggestions for Authors
In my opinion, in its actual form, the paper can be accepted for publication.
Author Response
The authors would like to thank the following for performing the review
Reviewer 2 Report
Comments and Suggestions for Authors
After thoroughly reviewing the revised manuscript and considering the authors' responses, I find the manuscript remains weak, particularly due to the lack of clear methodological development, which is crucial for scientific rigor. The authors have not adequately addressed our comments.
Introduction
The introduction fails to sufficiently summarize the current state of the field or clearly define the study’s aim. The cited literature does not appropriately support the research objectives and should be replaced with more relevant sources. The role of butyric acid (SCFAs) and sodium butyrate ( salt) remains undefined and insufficiently explained.
Methods
The experimental design is flawed, and the methods are poorly described, lacking sufficient detail for replication or proper evaluation. The approach to lactic acid determination is incorrect, leading to contradictory results.
It is impossible for the lactic acid produced by the bacteria during fermentation to be between 0.76 and 0.96 g/liter, (i.e. 0.076 to 0.096% lactic acid), when, as presented in table 3 of the manuscript, the lowest pH values (4.23) and the highest pH (4.63)
Results and Discussion
The results and discussion sections lack adequate justification for the findings (pH, acidity, color, etc.), and the references cited are insufficient.
Given these shortcomings, I do not recommend the manuscript for publication.
Author Response
We gratefully acknowledge your constructive comments and the time you have dedicated to evaluating our manuscript. In response, we have implemented your suggestions throughout the text, and all revisions are clearly indicated.
Introduction
The introduction fails to sufficiently summarize the current state of the field or clearly define the study’s aim. The cited literature does not appropriately support the research objectives and should be replaced with more relevant sources. The role of butyric acid (SCFAs) and sodium butyrate ( salt) remains undefined and insufficiently explained.
Thank you for your constructive feedback. We have revised the Introduction (L48-68; L119-128; L152-172). These changes better define the study’s aim and strengthen literature support. We trust the revised Introduction now meets the journal’s standards.
Methods
The experimental design is flawed, and the methods are poorly described, lacking sufficient detail for replication or proper evaluation. The approach to lactic acid determination is incorrect, leading to contradictory results.
Thank you for your insightful comment. In our field it is customary to reference established protocols rather than reproduce every procedural step. However, to ensure full reproducibility and in response to your feedback, we have expanded the Methods section to include some critical parameters.
L212, L231-234, L236-242, L280-281, L293-294
It is impossible for the lactic acid produced by the bacteria during fermentation to be between 0.76 and 0.96 g/liter, (i.e. 0.076 to 0.096% lactic acid), when, as presented in table 3 of the manuscript, the lowest pH values (4.23) and the highest pH (4.63)
Thank you for this valuable comment. We recognize that our lactic acid results were mistakenly reported as grams per liter, rather than as percent lactic acid. To correct this, we have updated the Methods section to use the normalized titratable acidity formula (L216).
All lactic acid values have been recalculated and are now expressed as percent lactic acid, consistent with the observed pH range. We trust these revisions fully address the concern.
Our revised protocol specifies:
- Accurately weigh 25 g of fermented milk into the titration flask.
- Add phenolphthalein indicator (0,5 mL) and swirl gently to ensure homogeneity and clear endpoint visualization.
- Titrate with 0.1 N NaOH to a persistent faint pink endpoint.
Results and Discussion
The results and discussion sections lack adequate justification for the findings (pH, acidity, color, etc.), and the references cited are insufficient.
Thank you for your comment. In the Results and Discussion section, we have made revisions and expanded the discussion— all changes have been clearly marked in the text (L387-388, L392-397, L399-428, L439-454, L476-480, L517-528, L541-544, L553-559, L594-599,L674-681).
Round 3
Reviewer 2 Report
Comments and Suggestions for Authors
Following both revisions, we found that the manuscript has improved, particularly in fundamental sections such as the Materials and Methods and the Introduction. However, to enhance the credibility and contextual grounding of the manuscript, I recommend that the authors establish a more thorough and well-documented correlation between the cited references and the content presented in key sections—particularly the Introduction and Results/Discussion. Strengthening these connections will significantly contribute to the manuscript’s scholarly rigor and overall relevance.
In conclusion, if the above suggestions are adequately addressed, I would be able to recommend the manuscript for publication.